# SIC-GAN: A Self-Improving Collaborative GAN for Decoding Sketch RNNs

## Abstract

Variational RNNs are proposed to output "creative" sequences. Ideally, a collection of sequences produced by a variational RNN should be of both high quality and high variety. However, existing decoders for variational RNNs suffer from a trade-off between quality and variety. In this paper, we seek to learn a variational RNN that decodes high-quality *and* high-variety sequences. We propose the *Self-Improving Collaborative GAN* (SIC-GAN), where there are two generators (variational RNNs) collaborating with each other to output a sequence and aiming to trick the discriminator into believing the sequence is of good quality. By deliberately *weakening* one generator, we can make another *stronger* in balancing quality and variety. We conduct experiments using the QuickDraw dataset and the results demonstrate the effectiveness of SIC-GAN empirically.

## 1 Introduction

Recurrent Neural Networks (RNNs) are popular models designed for sequence prediction problems. They have made significant impact on tasks related to Natural Language Processing (NLP), speech recognition, image captioning, and many other time-series analysis tasks. Traditionally, RNN decoding methods (Gu et al. (2017a); Boulanger-Lewandowski et al. (2013); Sutskever et al. (2014); Graves (2012); Cho (2016); Graves (2013); Gu et al. (2017b); Li et al. (2016)) focus on how to generate *high-quality* sequences. For example, in a language translation task, an RNN decoder aims to output a sequence that best expresses the input sequence in the target language. However, recent applications, such as Sketch drawing (Jongejan et al. (2016)), may expect an RNN to produce a *variety* of output sequences so to be "creative." Thus when drawing a firetruck, which is an output sequence of strokes, the RNN is expected to achieve both high quality (i.e., the sequence looks like a firetruck) and variety (i.e., different outputs looks different).

The above reasoning motivates the creation of *variational RNNs* that are able to output different sequences given the same input. There are two well-known ways to create the variety, namely *output sampling* (Graves (2013)) and *noise injection* (Chung et al. (2015); Cho (2016)), which are presented in Figures 1(b) and 1(c), respectively. For output sampling, to produce a point in a sequence, the decoder first outputs parameters of a predefined distribution, and then samples a point from the parametrized distribution. The process of sampling generates variety. While for *noise injection,* assuming that a small perturbation in the hidden space will not change a lot the semantic meaning in the input space, the decoder adds noises into the hidden representations $\boldsymbol{h}_i$ at each time step $i$, which in turn creates variety by changing the input slightly.

However, in practice, the variational RNNs usually generate outputs with low quality (Graves (2013)). Figure 2(a) shows some examples of output sequences, which are sketches of firetrucks, produced by a variational RNN using output sampling.[1] As we can see, many sequences are *not* recognizable as firetrucks. Similarly, the outputs of variational RNNs using noise injection also lack quality when the noise level is high.

To solve the problem, one can employ decoding techniques that improve the quality. Ha & Eck (2017) introduce a *temperature* parameter in the model that controls the level of randomness in output sampling (Figure 1(b)) or in hidden noises (Figure 1(c)). This concentrates around the mean of

---

[1]For more details about the settings and training process, please refer to Section 5.

the distribution of points that can be output at each time. In beam search (Graves (2012); Boulanger-Lewandowski et al. (2013); Sutskever et al. (2014)), the decoder saves several output points into a tree-structural buffer at each time step, and finds out from the tree the best combination of points (one point at a time step) that will constitute as the output sequence. In effect, this concentrates the distribution of the sequences to be output by an RNN to its mean. Unfortunately, these decoding methods, when applied to variational RNNs, create a trade-off between quality and variety. Figures 2(b) and 2(c) illustrate some examples of sequences produced using the temperature method and beam search method, respectively. It can be seen that the quality is improved but at the cost of limited variety.

Figure 2(d) displays human-drawn sketches of fire trucks which demonstrate that producing sequences–in this case sketches–with both quality and variety is achievable by humans. The challenge is to come up with an algorithm that can produce the same (if not better) quality and variety of outputs as humans can.

In this paper, we seek to learn variational RNNs that output high-quality and high-variety sequences. We found out that the reason why variational RNNs give low quality is mainly because of the *error propagation* effect—a "bad" point output at the current time step by randomness will create a negative impact on the future points. On the other hand, *it is the "bad" points that create variety*. So instead of avoiding such a bad point, as in most existing work try to do, our key idea is to allow a bad point to occur but with the expectation that its negative impact in the future may be "covered" by some mechanism later on.

In line with this, we propose the *Self-Improving Collaborative GAN* (SIC-GAN), where there are two generators (variational RNNs) collaborating with each other to generate an output sequence and aiming to trick the discriminator into believing the sequence is of good quality. We deliberately let one generator, which we call the *weak* generator, produce bad points. This forces another generator, we refer as the *strong* generator, to learn to cover the bad points created by its partner in order to successfully fool the discriminator. After training SIC-GAN, we keep the strong generator only. This final variational RNN (the strong generator) will have the ability to conceal inadvertently bad points made by itself, and will output sequences of both high quality and variety.

To the best of our knowledge, this is the first work that improved the outputs of variational RNN in terms of both quality and variety. Following summarizes our contributions:

- We propose a novel collaborative GAN model that guides a variational RNN to learn to cover the creative points and to preserve the quality of an output sequence.

- The proposed collaborative GAN opens up some new research directions. For example, how the collaboration is done? How to weaken one generator to make another strong? We provide initial steps toward these directions.

- The collaborative GAN has an interesting extension in that one can use a trained strong generator as the weak generator and train a "stronger" generator, and this process can be repeated until the generalizability of the discriminator is fully exploited. We call such an iterative process the *self-improving* training technique. While this technique is similar to the fictitious self-play that has been seen in an Reinforcement Learning agent such as AlphaGo (Silver et al. (2016)), we believe it is new to the GAN field.

- We conduct experiments using the QuickDraw dataset (Jongejan et al. (2016)) released by Google, and the results show that SIC-GAN achieves the state-of-the art performance in striking the balance between quality and variety.

## 2 RELATED WORK

There are some techniques other than the beam search and temperature method that improves the quality of RNN decoding. Li et al. (2016) observe that beam search decoding often sticks in a few branches. They modify the beam search algorithm to penalize siblings. This method helps find the high-quality sequences more efficiently, but it is still subject to the same problem as the beam search—there is no guarantee that the found sequences are diverse. Greedy decoding (Germann (2003); Langlais et al. (2007)) outputs a sequence by choosing the most likely point at each local

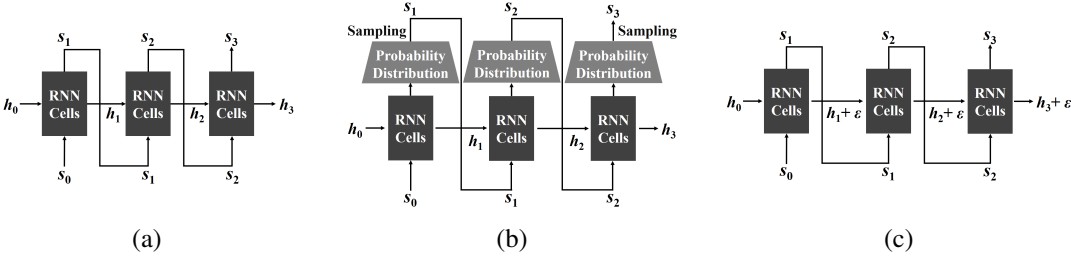

Figure 1: RNN models. (a) Vanilla RNNs. (b) Variational RNNs based on output sampling. (c) Variational RNNs based on noise injection.

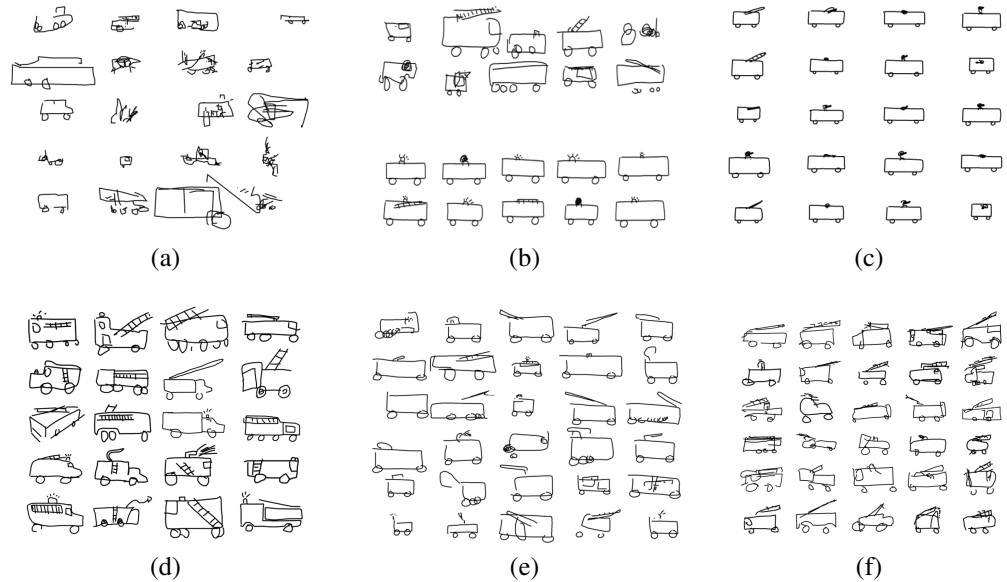

Figure 2: Firetrucks drawn by (a) variational RNN; (b) tempered variational RNN parameter (top: high temperature, bottom: low temperature); (c) variational RNN with beam search; (d) human beings; (e) variational RNN trained by Naive GAN; (f) variational RNN trained by SIC-GAN.

step. It is a special case of the temperature method or beam search (where the beam size equals 1), and subject to the same quality-variety trade-off.

Recently, studies (Yu et al. (2017); Che et al. (2017); Kusner & Hernández-Lobato (2016); Zhang et al. (2016)) have explored the idea of applying GANs to generate better sequences. In particular, there are some GAN-based approaches that focus on the RNN decoding. The professor forcing (Goyal et al. (2016)) uses GANs to reduce the mismatch between the training and testing distributions of the RNN output. Their goal is different from ours. The Gumbel-greedy decoding (Gu et al. (2017b)) uses GANs to fine-tune a variational RNN such that it outputs better-quality sequences. This provides a way other than the beam search to find high-quality sequences. But the work is still subject to the same quality-variety trade-off as the beam search.

Some studies use the Reinforcement Learning to improve the RNN decoding. SeqGAN (Yu et al. (2017)) and the work by Bahdanau et al. Bahdanau et al. (2016) uses a discriminator to guide the RNN decoding based on the policy gradients. The goal is to solve the gradient passing problem in GANs on discrete output and is different from that of this paper. Trainable Greedy Decoding (TGD) (Gu et al. (2017a)) extends the idea of noise-injected variational RNNs (Cho (2016)) by formulating the decoding as a Reinforcement Learning problem. TGD learns an agent to decide what noise to inject, and directly uses the BLEU score (Papineni et al. (2002)) as its reward function to guide the agent. Note that BLEU is a quality measure thus TGD improves the quality of the output sequences only. To improve the variety simultaneously, one has to define a reward function for the variety.

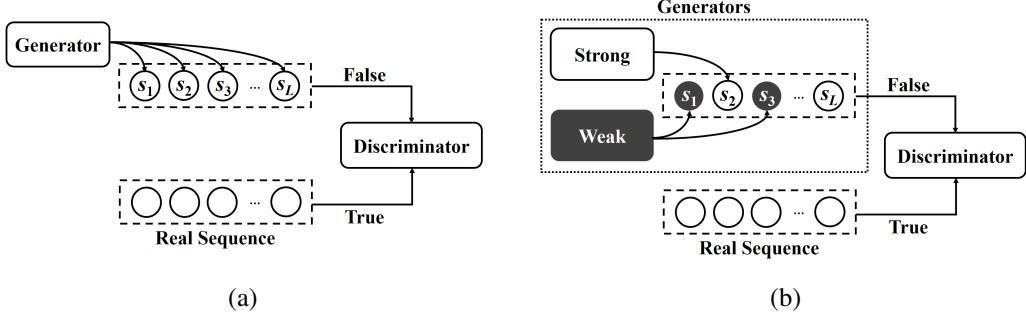

Figure 3: Model architecture. (a) Naive GAN. (b) The Strong-Week Collaborative GAN.

Unfortunately, there is no existing study about such a reward function. SIC-GAN differs from the Reinforcement Learning-based approaches in that it does *not* require any variety reward/measure to train, which we believe is a significant advantage for applications (such as drawing and image data synthesis tasks) whose output is in a high dimensional space and/or in an unknown manifold of that space.

## 3   STRONG-WEAK COLLABORATIVE GAN

In this section, we introduce a novel GAN setting, called the *strong-weak collaborative GAN*, which our SIC-GAN is based on. We then present SIC-GAN in Section 4.

The output of vanilla variational RNNs lacks quality because of the error propagation; that is, a creative point generated at a time step for variety prevents the points at later time steps from forming a high quality sequence. Most existing decoding methods improve the quality of the output sequences by reducing the variation of points, but this creates a trade-off between the quality and variety. To avoid such a trade-off, we do *not* limit the variation of a point. Instead, we let the later points learn to *cover* the negative effect of the variation.

To achieve our goal, one naive way is to use a GAN to improve a variational RNN, as shown in Figure 3(a). Given a (pre-trained) variational RNN, we can use it as the generator in GAN that output sequences aiming to trick the discriminator into believing they are real. In the presence of variation of a point $s_i$ at time step $i$, the generator would learn to cover the error (in terms of quality) of $s_i$ when generating later points $s_j$'s, $j > i$, during the GAN training process in order to successfully fool the discriminator. Note that the discriminator performs the discrimination *at the sequence level*, so the generator has a chance to output high quality sequences without sacrificing the variety of each generated point. Figure 2(e) shows some example sequences output by this naive GAN approach. We can see that it improves *both* the quality and variety of the output sequences to a certain degree.

However, the naive GAN approach has a subtle drawback—it does *not* always preserve the variety of a variational RNN. This is because that, during the GAN training, there is another way for the generator to fool the discriminator. At each time step, the generator can choose to reduce the variation and output a more probable point that leads to a high quality sequence in order to fool the discriminator easily. In this case, the variety is limited.

To avoid such a problem, we propose the strong-weak collaborative GAN where there are strong and weak generators *collaborate* to generate an output sequence aiming to fool the discriminator, as shown in Figure 3(b). We deliberately *weaken* the weak generator by letting it output "bad" (variational) points. Thus, the strong generator is forced to learn to cover the variation during the GAN training process. After the training, we discard the weak generator and replace it by the strong one to get the final variational RNN. This RNN, after being weakened, can output sequences of both quality and variety. Following the improved WGAN (Gulrajani et al. (2017)), the objective for the strong generator $G^+$, weak generator $G^-$, and discriminator $D$ is:

$$\arg \min_{\theta^+, \theta^-} \max_{\phi} \mathbb{E}_{\mathbf{x} \sim P_{\text{data}}}[D(\mathbf{x}; \phi)] - \mathbb{E}_{\mathbf{z}^+, \mathbf{z}^- \sim \mathcal{N}}[D(G^+(\mathbf{z}^+; \theta^+) \odot G^-(\mathbf{z}^-; \theta^-); \phi)],$$

where $P_{data}$ is the real data distribution and $\odot$ denotes the collaboration between $G^+$ and $G^-$. We omit the gradient penalty term for simplicity.

The strong-weak collaborative GAN is a general architecture and there are different ways to define the collaboration and weakening. For the collaboration, one can let the weak generator generate the first half of a sequence and then let the strong one generate the second half. Here, we choose to let the two generators take turns to generate points in a sequence, so the strong generator can see the "bad" points right after they are generated and learn to take actions to recover the quality immediately. The detailed architecture of our collaboration design is shown in Figure 4(a). As compared to standard RNNs, the collaborative version has roughly the same number of weights to update in an unfolded .

For the weakening, we simply raise the temperature Ha & Eck (2017) of the weak generator such that a generated (partial) sequence less concentrates around its mean. We fix the weights of the weak generator during the GAN training process and update the weights of the strong generator only. Since the weak generator is not updated and could generate "bad" points from time to time by its nature, the strong generator is now forced to learn to cover the negative impact of the variety of input variational RNN in order to successfully fool the discriminator.

Note that one may wish to just weaken the ordinary RNN generator of the naive GAN described above to achieve the "covering" effect similar to that of the strong-weak collaborative GAN—if a point is made bad, the RNN may learn to generate better points at later time steps in order to fool the discriminator. However, the "next points" in the weakened RNN are made bad too (since the entire RNN is weakened) and may not be able to actually cover the previous point. To fool the discriminator in such a situation, the RNN may instead learn to output points that, after being weakened, are more easily "covered" by the future (bad) points. In effect, this makes the RNN conservative to generating novel sequence and reduces variety. We call this the *covering-or-covered paradox*.

## 4   SIC-GAN: SELF-IMPROVING COLLABORATIVE GAN

Once trained, the strong generator in the strong-weak collaborative GAN is used to generate an entire sequence. This means that the strong generator should have enough based temperature (or noise level) to ensure the variety. One naive way to do so is to add a base-temperature to both the strong and weak generators during the training time. However, the strong generator faces the covering-or-covered paradox now and may learn to be conservative.

We can instead train the strong-weak collaborative GAN multiple times using a *self-improving* technique, as shown in Figure 4(b). We start by adding a low base-temperature to both the strong and weak generators and train them in the first phase. Then we set the weak generator in the next phase as the strong one we get from the previous phase and train the generators with increased base-temperature. We then repeat this process until the target base-temperature is reached. We call the process "self-improving" because the strong generator in the next phase learns to cover itself in the previous phase. It is important to note that in a latter phase, the weak generator is capable of covering the negative effect due to the variety of the strong generator (because that weak generator is a strong generator in the previous phase). So, the strong generator in the current phase can focus on the "covering" rather than "covered," preventing the final RNN from being conservative.

Although viable, training a long series of GANs could be time consuming. Here, we propose the *Self-Improving Collaborative GAN* (SIC-GAN) that improves the strong generator in a single GAN training process. The idea is to simply *tie the weights of the strong and weak generators together*. Note that the tying is done in the soft manner; that is, we add a loss term

$$\|\theta^- - \theta^+\|^2$$

for $G^-$. Every time the strong generator improves, the improvement is immediately propagated to the weak generator. This allows the strong generator to learn to cover the quality errors made by itself over the training iterations.

In addition to time efficiency, the SIC-GAN offers some advantages over the strong-weak collaborative GAN. First, the self-improving ability could lead the strong generator to *learn beyond the knowledge* provided by the training dataset up to the generalizability of the discriminator in striking the balance between quality and variety. The self-improving ability has been discussed in some

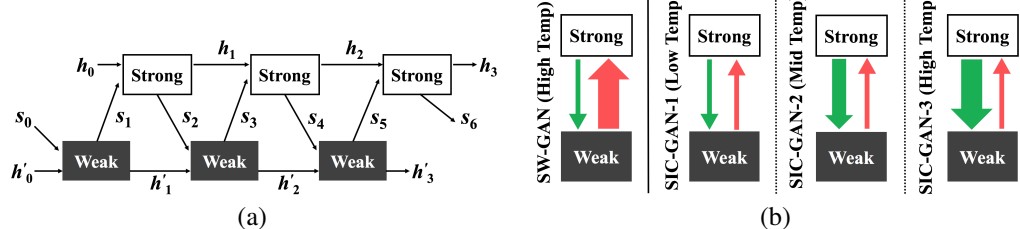

Figure 4: (a) Collaboration between the strong and weak generators for outputting a sequence. (b) Self-improving technique for coping with the covering-or-covered paradox. The green arrows denote the degrees of covering effect learned by the strong generator; while the red arrows denote the degrees of covered effect learned by the strong generator.

Reinforcement Leaning work such as the AlphaGo (Silver et al. (2016; 2017)). By playing Go with and beating itself iteratively, the agent is able to learn what was *not* in the playbooks written by humans. Note that the SIC-GAN requires only the measure of quality (used by the discriminator) while a Reinforcement Learning-based agent would require the reward functions of both quality and variety to train.

## 5   EXPERIMENTS

Next, we conduct experiments using the QuickDraw (Jongejan et al. (2016)) dataset and evaluate the effectiveness of SIC-GAN in terms of both quality and variety.

The QuickDraw dataset[2] consists of 50 million hand-drawn sketches in 345 classes. Each sketch in turn consists of multiple pen strokes. A stroke is recorded in the form $(\Delta x, \Delta y, p_1, p_2, p_3)$, where $(\Delta x, \Delta y)$ denotes the ending position of the stroke relative to that of the previous stroke, and $(p_1, p_2, p_3)$ represents a one-hot vector of 3 possible states. The state $p_1$ indicates if the pen is currently touching the canvas. The state $p_2$ tells if the pen will be lifted from the canvas after the current stroke. While the state $p_3$ indicates whether the sketch drawing has ended.

**Settings.** To set up a GAN, we use a CNN consisting of a 1D convolution layer, a max-pooling layer, and a fully-connected output layer as the discriminator. In the convolution layer, we use the filters proposed by Kim (2014) that scan the $n$-grams of the real and generated sequences. The layer scans 1- to 30-grams, each with a 128 filters. The generator is a variational RNN consisting of 1-layer LSTM with 512 hidden units. At each time step, the generator outputs parameters of a mixture of 20 two-dimensional Gaussian distributions and also parameters of a Categorical distribution of three categories. The parametrized Gaussian mixture model is then used to sample the $(\Delta x, \Delta y)$ part of a generated point, and the parametrized Categorical distribution is used to sample the $(p_1, p_2, p_3)$ part. We can gradually increase the base-temperature during the training process by following the *curriculum learning* strategy (Bengio et al. (2009; 2015)). We use SIC-GAN to improve the decoding of a variational RNN pre-trained by labeled examples aiming to draw a sketch of firetruck given an initial stroke. We use the improved WGAN (Gulrajani et al. (2017)) as the training algorithm. We update the generator every batch and the discriminator every 5 batches. We implement SIC-GAN using TensorFlow 1.2 and our code is based on that of the *sketch rnn* (Ha & Eck (2017)).[3]

**Baseline Methods.** We compare SIC-GAN against the several baseline decoding methods, including the ordinary decoder (Vanilla) of a variational RNN using output sampling (Graves (2013)), the decoder with a lowered temperature (Low Temp) by Ha & Eck (2017) where the temperature parameter is set to $0.2$, beam search (Beam) by Graves (2012); Boulanger-Lewandowski et al. (2013); Sutskever et al. (2014) which samples 10 points and keep the top-5 points ranked by their respective probability at each time step, and the naive GAN (GAN) described in Section 3. Note that the SIC-GAN has a plausible simplification that uses only *one* generator with a random noise $\delta_i$ added to the output point at every one another time step, $i = 1, 3, 5, \cdots$. We also implement this approach (Noisy GAN) and use it as a baseline.

---

[2]https://github.com/googlecreativelab/quickdraw-dataset
[3]https://github.com/tensorflow/magenta/tree/master/magenta/models/sketch_rnn

## 5.1 QUALITY

**Measures.** To evaluate the quality of the sequences produced by a variational RNN, we need to have quality measures first. However, there is no existing quality measure defined in the literature for sketch drawing. Here, we propose two new measures, namely the *typicalness* and *details*, for this task. We find that the strokes in a sketch can usually be divided into two groups serving different purposes. One contains the strokes for the main contour and the other includes the strokes for the details. For example, when drawing a firetruck, we can usually see strokes for the body and wheels, as shown in Figure 5(a), that defines the main contour. We can also see strokes for the bell and ladder, as shown in Figure 5(b), that adds details. To quantify the typicalness, we train a CNN with 4 convolution layers and 4 fully connected layers that classifies 5 carefully chosen classes—the airplanes, bicycles, firetrucks, helicopters, and submarines. These classes define very different main contours, so the classifier tends to capture the difference between the main contours of the input sequences when making predictions. We use the CNN to obtain a *typicalness score* of outputs of a variational RNN drawing firetrucks. Given 1000 sequences generated by the RNN, the typicalness score is defined as the times the CNN classifies the sequences into the firetruck class. Similarly, for quantifying the details measure, we train another CNN of the same architecture using the examples in 5 carefully chosen classes, namely the ambulances, buses, firetrucks, pickup trucks, and police cars. This time, sketches in these classes share the main contours but are different from each other in details. Therefore, the classifier tends to distinguish the details of input sequences when making predictions. Given 1000 sequences generated by the RNN, the *details score* is defined as the times the CNN classifies the sequences into the firetruck class. Note that the above two CNNs are trained using the rectified images (in pixels) so to ensure that the scores are derived purely from the appearance of the sketches.

**Base variety.** To see how different decoding approaches balance quality and variety, we evaluate the typicalness and details scores at different levels of *base variety*. We create the base variety at degree $1/i$ by replacing the points with random noises at every $i$ time steps when a model is generating a sequence at the test time. Here, we take $i = 3, 5, 10$ into account.

| | Vanilla | Low Temp | Beam | GAN | Noisy GAN | SIC-GAN |
|---|---|---|---|---|---|---|
| Typicalness@0 | 0.995 | **0.998** | 0.973 | 0.993 | 0.99 | 0.986 |
| Details@0 | 0.793 | 0.813 | 0.546 | 0.854 | **0.885** | 0.845 |
| Typicalness@(1/10) | 0.872 | 0.808 | 0.901 | 0.897 | 0.869 | **0.913** |
| Details@(1/10) | 0.573 | 0.483 | 0.508 | 0.78 | 0.773 | **0.784** |
| Typicalness@(1/5) | 0.644 | 0.557 | 0.804 | 0.681 | 0.738 | **0.823** |
| Details@(1/5) | 0.495 | 0.388 | 0.442 | 0.669 | 0.686 | **0.701** |
| Typicalness@(1/3) | 0.425 | 0.428 | **0.706** | 0.537 | 0.529 | 0.651 |
| Details@(1/3) | 0.539 | 0.474 | 0.440 | 0.609 | 0.609 | **0.664** |

Table 1: Typicalness and details scores at (@) different base variety.

**Results.** Table 1 lists the typicalness and details scores of different approaches at different degrees of base variety. When there is no base variety, all the models perform roughly the same. The only exception is Beam, which gives a particularly low details score. This implies that it can only generate firetrucks of typical contours, as we have seen in Figure 2(c). As the degree of base variety increases, SIC-GAN starts to outperform other methods in *both* typicalness and details. Figures 6 also shows that some non-GAN-based methods tend to confuse the firetrucks with some other classes. Figure 2(f) shows the firetrucks drawn by SIC-GAN. We can clearly see that the firetrucks contains more details than those output by GAN. Note that the Noisy GAN outperforms SIG-GAN when the base variety is low, but degrades dramatically when the base variety goes up. The two-generator design is advantageous for coping with the covering-or-covered paradox.

## 5.2 VARIETY

We quantify the variety of a model by directly calculating the sample variance of 100 output sequences. To calculate the sample variance, we need a distance measure between the sketches. One way is to use the 1-norm, as shown in Figure 7(a), where the distance is the sum of differences

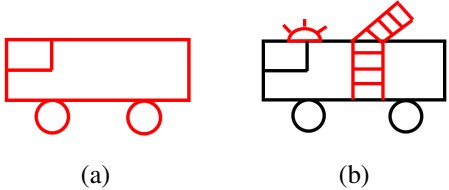

Figure 5: Two type of strokes. (a) Main contour. (b) Details.

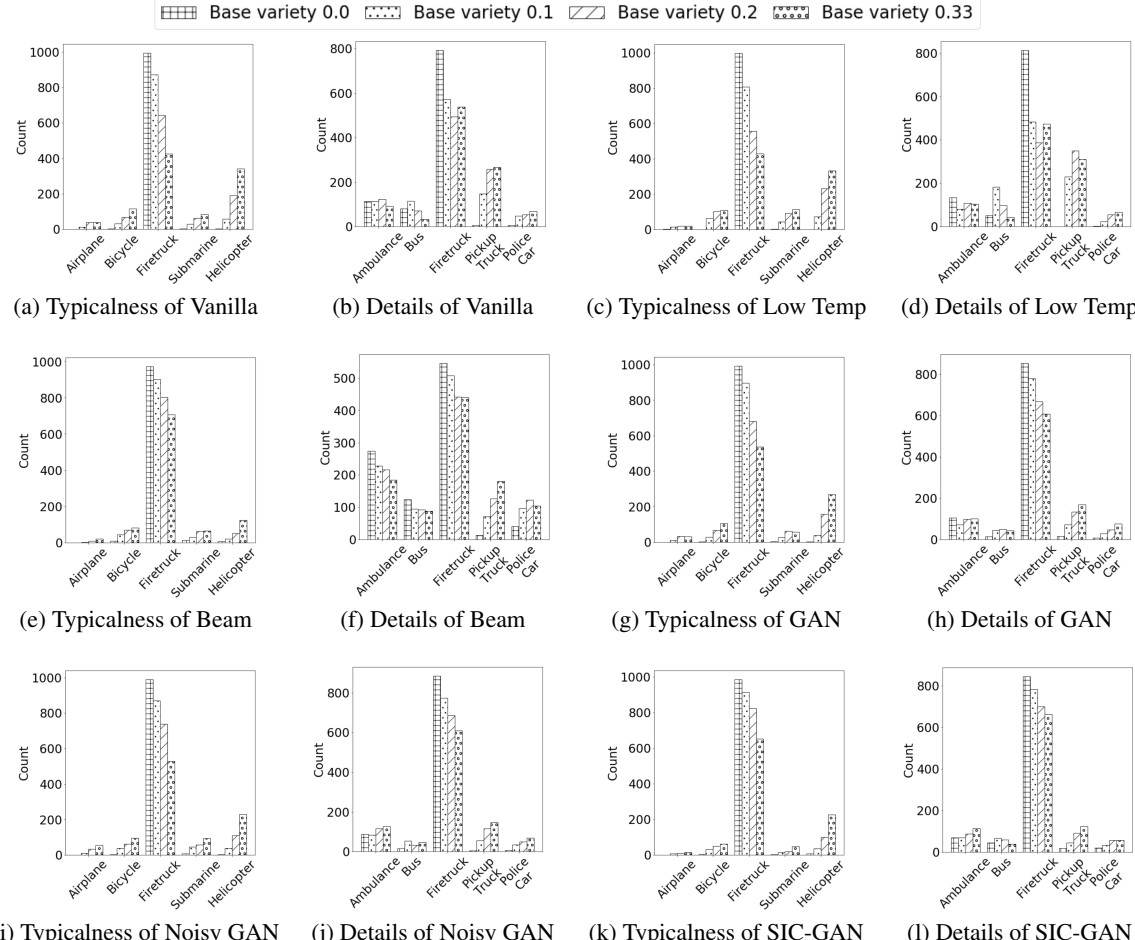

(a) Typicalness of Vanilla    (b) Details of Vanilla    (c) Typicalness of Low Temp    (d) Details of Low Temp

(e) Typicalness of Beam    (f) Details of Beam    (g) Typicalness of GAN    (h) Details of GAN

(i) Typicalness of Noisy GAN    (j) Details of Noisy GAN    (k) Typicalness of SIC-GAN    (l) Details of SIC-GAN

Figure 6: Distributions of the predictions of typicalness and details CNNs over 1000 output sequences.

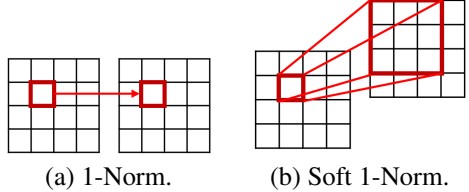

(a) 1-Norm.      (b) Soft 1-Norm.

Figure 7: Difference between standard and soft 1-norm.

|  | Vanilla | Low Temp | Beam | GAN | Noisy GAN | SIC-GAN |
|---|---|---|---|---|---|---|
| 1-Norm | 135.88 | 115.18 | 109.01 | 122.29 | 115.56 | **145.28** |
| Gaussian Blur + 1-Norm | 43.33 | 37.02 | 35.88 | 36.76 | 34.11 | **48.81** |
| Soft 1-Norm | 3.99 | 1.38 | 2.30 | 4.59 | 4.14 | **6.43** |

Table 2: Variety scores given different similarity measures.

between the RGB values of the pixels in rectified images. Note that the standard 1-norm does not take into account the translation, rotation, scaling, and other perturbations of an image that would not change the semantic distance. So, we walk around this problem by either adding a Gaussian blur to the rectified sketches first or using the soft 1-norm, i.e., the sum of the differences between the RGB values of the nearby pixels in rectified images, as shown in Figure 7(b). Table 2 shows the variety scores achieved by different approaches. SIC-GAN consistently gives the highest variety in all measures. This can be seen more specifically in Figure 2(f), where SIC-GAN generates more diverse results than GAN. Furthermore, some of the firetrucks it generates do not bear a resemblance to any training example, implying that SIC-GAN can learn beyond the provided knowledge.

## 6 CONCLUSIONS

We propose the strong-weak collaborative GAN and SIC-GAN that fine-tune a variational RNN to the one capable of generating high-quality and high-variety sequences. We conduct experiments using a real-world dataset and the results shows that SIC-GAN can achieve the state-of-the-art performance in balancing quality and variety. These work opens up several interesting research directions. For example, as the strong-weak collaborative GAN is a general architecture, there could be other ways to define the collaboration between the strong and weak generators as well as the weakening mechanism for the weak generator. Also, it is possible to apply the SIC-GAN to other applications such as the Natural Language Processing (NLP) tasks. These are the matters of our future exploration.

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

## APPENDIX

To help understand the distribution of the output sequences of different methods, we include more generated sketches here.

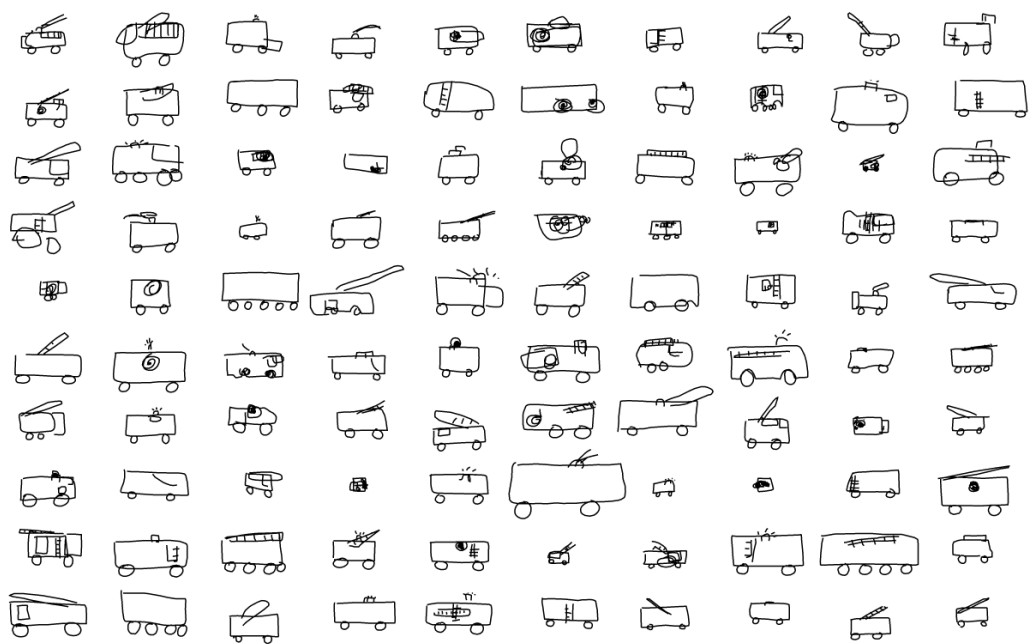

Figure 8: Firetrucks by Vanilla.

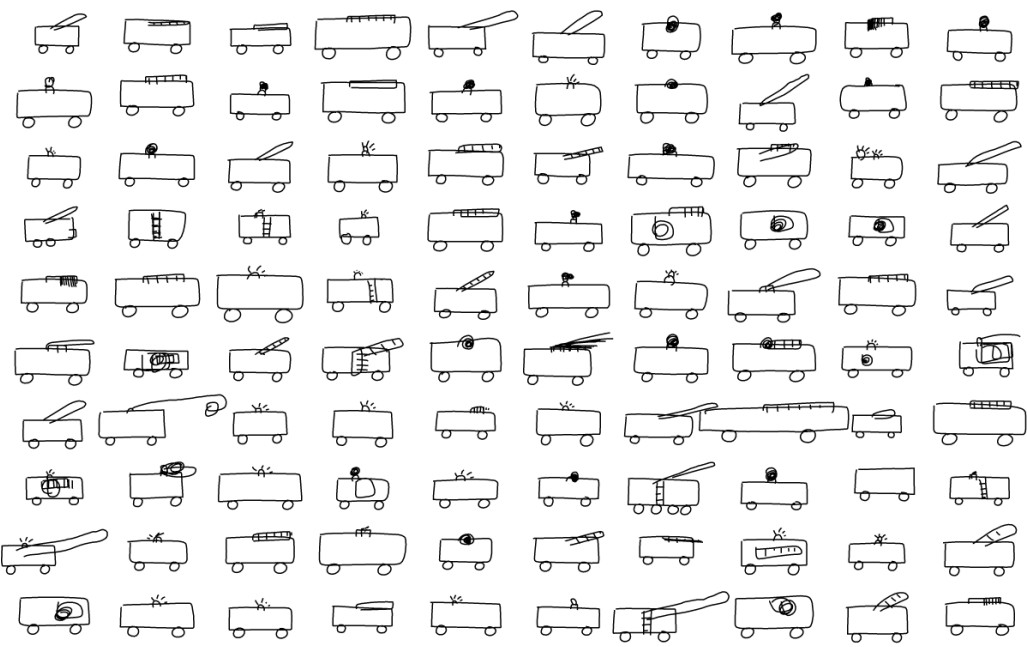

Figure 9: Firetrucks by Low Temp.

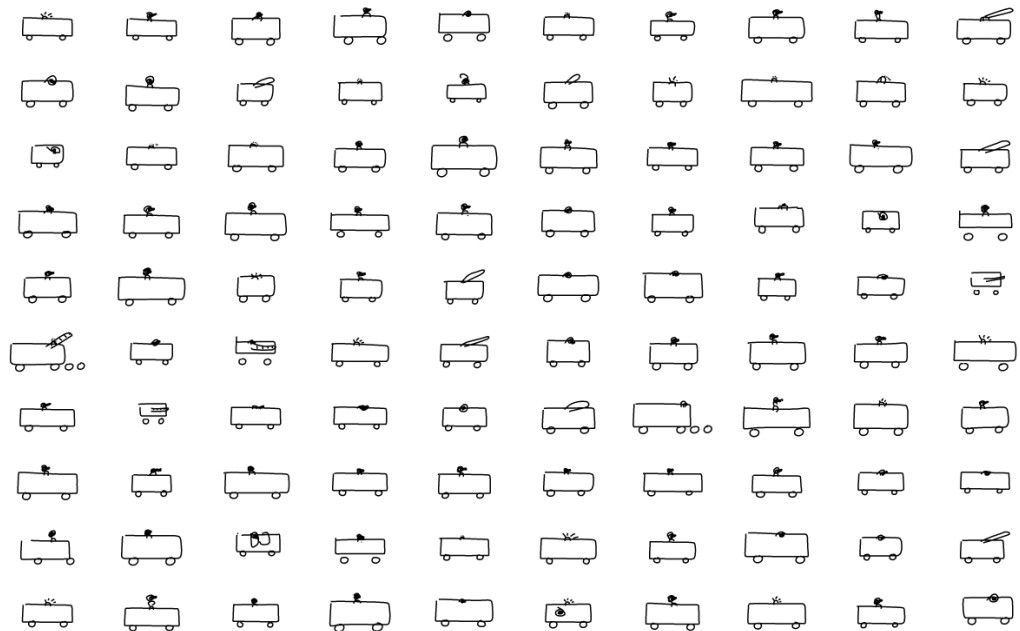

Figure 10: Firetrucks by Beam.

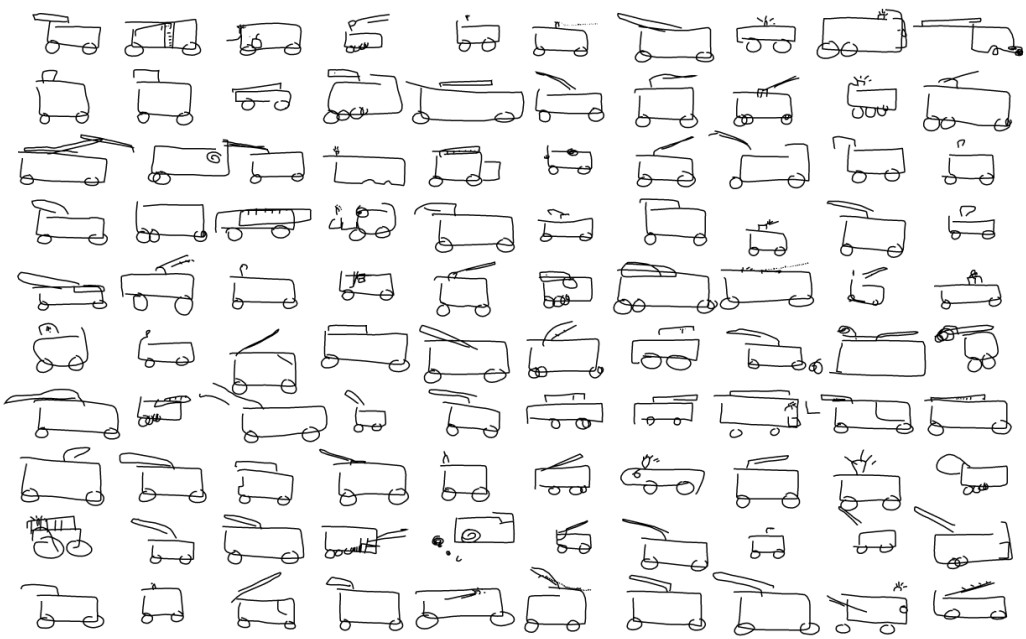

Figure 11: Firetrucks by GAN.

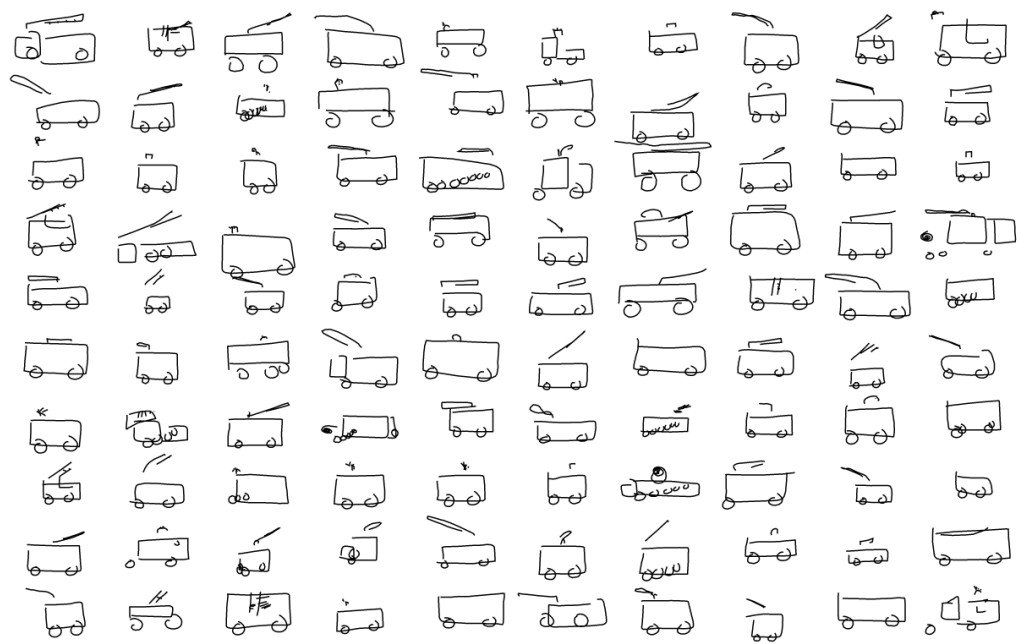

Figure 12: Firetrucks by Noisy GAN.

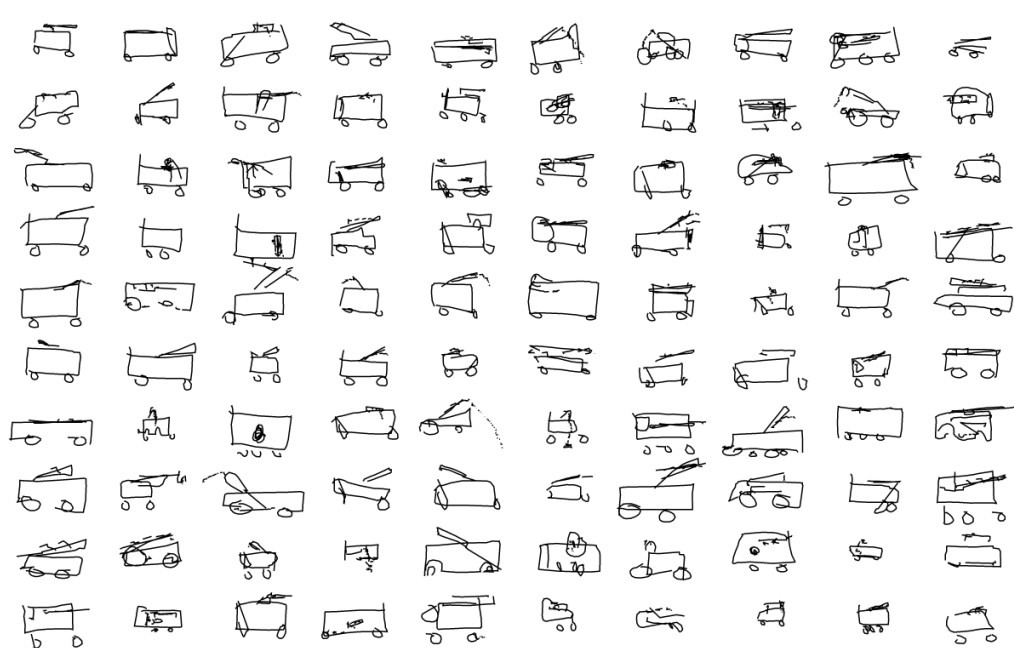

Figure 13: Firetrucks by SIC-GAN.

