# OpenReview forum: "SIC-GAN: A Self-Improving Collaborative GAN for Decoding Sketch RNNs"
_ICLR.cc/2018/Conference — Reject_

### Official Review · AnonReviewer3 · 2017-11-27
**Interesting idea, but requires more datasets to show.**

**Rating:** 5
**Confidence:** 3

**Review:**

The paper proposed a method that tries to generate both accurate and diverse samples from RNNs.
I like the basic intuition of this paper, i.e., using mistakes for creativity and refining on top of it. I also think the evaluation is done properly. I think my biggest concern is that the method was only tested on a single dataset hence it is not convincing enough. Also on this particular dataset, the method does not seem to strongly dominate the other methods. Hence it's not clear how much better this method is compared to previously proposed ones.

---

> ### Author Response · Authors · 2018-01-05
> **Thanks.**
>
> We thank the reviewer for constructive comments.
>
> Q: I think my biggest concern is that the method was only tested on a single dataset...
> A: Thanks. Following the suggestion of reviewer 2, we have changed the paper title to Sketch RNN so we believe this is no longer a concern.

---

### Official Review · AnonReviewer2 · 2017-11-27
**A novel architecture for generating greater variety on QuickDraw dataset, but seems confused about what it's actually doing.**

**Rating:** 4
**Confidence:** 5

**Review:**

This paper baffles me. It appears to be a stochastic RNN with skip connections (so it's conditioned on the last two states rather than last one) trained by an adversarial objective (which is no small feat to make work for sequential tasks) with results shown on the firetruck category of the QuickDraw dataset. Yet the authors claim significantly more importance for the work than I think it merits.

First, there is nothing variational about their variational RNN. They seem to use the term to be equivalent to "stochastic", "probabilistic" or "noisy" rather than having anything to do with optimizing a variational bound. To strike the right balance between pretension and accuracy, I would suggest substituting the word "stochastic"  everywhere "variational" is used.

Second, there is nothing self-improving or collaborative about their self-improving collaborative GAN. Once the architecture is chosen to share the weights between the weak and strong generator, the only difference between the two is that the weak generator has greater noise at the output. In this sense the architecture should really be seen as a single model with different noise levels at alternating steps. In this sense, I am not entirely clear on what the difference is between the SIC-GAN and their noisy GAN baseline - presumably the only difference is that the noisy GAN is conditioned on a single timestep instead of two at a time? The claim that these models are somehow "self-improving" baffles me as well - all machine learning models are self-improving, that is the point of learning. The authors make a comparison to AlphaGo Zero's use of self-play, but here the weak and strong generators are on the same side of the game, and because there are no game rules provided beyond "reproduce the training set", there is no possibility of discovery beyond what is human-provided, contrary to the authors' claim.

Third, the total absence of mathematical notation made it hard in places to follow exactly what the models were doing. While there are plenty of papers explaining the GAN framework to a novice, at least some clear description of the baseline architectures would be appreciated (for instance, a clearer explanation of how the SIC-GAN differs from the noisy GAN). Also the description of the soft $\ell_1$ loss (which the authors call the "1-loss" for some reason) would benefit from a clearer mathematical exposition.

Fourth, the experiments seem too focused on the firetruck category of the QuickDraw dataset. As it was the only example shown, it's difficult to evaluate their claim that this is a general method for improving variety without sacrificing quality. Their chosen metrics for variety and detail are somewhat subjective, as they depend on the fact that some categories in the QuickDraw dataset resemble firetrucks in the fine detail while others resemble firetrucks in outline. This is not a generalizable metric. Human evaluation of the relative quality and variety would likely suffice.

Lastly, the entire section on the strong-weak collaborative GAN seems to add nothing. They describe an entire training regiment for the model, yet never provide any actual experimental results using that model, so the entire section seems only to motivate the SIC-GAN which, again, seems like a fairly ordinary architectural extension to GANs with RNN generators.

The results presented on QuickDraw do seem nice, and to the best of my knowledge it is the first (or at least best) applications of GANs to QuickDraw - if they refocused the paper on GAN architectures for sketching and provided more generalizable metrics of quality and variety it could be made into a good paper.

---

> ### Author Response · Authors · 2018-01-05
> **Thanks.**
>
> We thank the reviewer for constructive comments. Following is our reply:
>
> Q: Once the architecture is chosen to share the weights between the weak and strong generator, ...it appears to be a stochastic RNN with skip connections (so it's conditioned on the last two states rather than last one) trained by an adversarial objective...
> A: We are sorry for not describing the “tying” precisely. It is done in a soft manner; that is, we add a loss term for the weak generator that require its parameters to be similar to those of the strong generator. Please see Section 4 for more details. Actually, the extra input taken by the strong generator is not necessary and are not implemented. We just described it for the cases when the hyperparameter of the term is high. We have remove the irrelevant sentences to avoid confusion.
>
> Q: I am not entirely clear on what the difference is between the SIC-GAN and their noisy GAN baseline...
> A: The noisy GAN just weakens the ordinary RNN generator of the naive GAN to achieve the “covering” effect similar to that of the strong-weak collaborative GAN—if a point is made bad, the RNN may learn to generate better points at later time steps in order to fool the discriminator. However, the “next points” in the weakened RNN are made bad too (since the entire RNN is weakened) and may not be able to actually cover the previous point. To fool the discriminator in such a situation, the RNN may instead learn to output points that, after being weakened, are more easily “covered” by the future (bad) points. In effect, this makes the RNN conservative to generating novel sequence and reduces variety. We call this the covering-or-covered paradox. On the other hand, once trained, the strong generator in the strong-weak collaborative GAN is used to generate an entire sequence. This means that the strong generator should have enough based temperature (or noise level) to ensure the variety. One naive way to do so is to add a base-temperature to both the strong and weak generators during the training time. However, the strong generator faces the covering-or-covered paradox now and may learn to be conservative. We can instead train the strong-weak collaborative GAN multiple times using a self-improving technique. We start by adding a low base-temperature to both the strong and weak generators and train them in the first phase. Then we set the weak generator in the next phase as the strong one we get from the previous phase and train the generators with increased base-temperature. We then repeat this process until the target base-temperature is reached. We call the process “self-improving” because the strong generator in the next phase learns to cover itself in the previous phase. It is important to note that in a later phase, the weak generator is capable of covering the negative effect due to the variety of the strong generator (because that weak generator is a strong generator in the previous phase). So, the strong generator in the current phase can focus on the “covering” rather than “covered,” preventing the final RNN from being conservative.
>
> Q: ...because there are no game rules provided beyond “reproduce the training set,” there is no possibility of discovery beyond what is human-provided, contrary to the authors' claim.
> A: The generator can exploit up to the generalizability of the generator.

---

### Official Review · AnonReviewer1 · 2017-11-27
**Very interesting approach for training a sequential generator adversarially**

**Rating:** 7
**Confidence:** 3

**Review:**

Overall the paper is good: good motivation, insight, the model makes sense, and the experiments / results are convincing. I would like to see some evidence though that the strong generator is doing exactly what is advertised: that it’s learning to clean up the mistakes from variation. Can we have some sort of empirical analysis that what you say is true?

The writing grammar quality fluctuates. Please clean up.

Detailed notes
P1:
Why did you pass on calling it Self-improving collaborative adversarial learning (SICAL)?
I’m very surprised you don’t mention VAE RNN here (Chung et al 2015) along with other models that leverage an approximate posterior model of some sort.

P2:
What about scheduled sampling?
Is the quality really better? How do you quantify that? To me the ones at the bottom of 2(c) are of both lower quality *and* diversity.
“Figure 2(d) displays human-drawn sketches of fire trucks which demonstrate that producing sequences–in this case sketches–with both quality and variety is definitely achievable in real-world applications”: I’m not sure I follow this argument. Because people can do it, ML should be able to?

P3:
“Recently, studies start to apply GANs to generate the sequential output”: fix this
Grammar takes a brief nose-dive around here, making it a little harder to read.
Caption: “bean search”
Che et al also uses something close to Reinforcement learning for discrete sequences.
“nose-injected”: now you’re just being silly
Maybe cite Bahdanau et al 2016 “An actor-critic algorithm for sequence prediction”
“does not require any variety reward/measure to train” What about the discriminator score (MaliGAN / SeqGAN)? Could this be a simultaneous variety + quality reward signal? If the generator is either of poor-quality or has low variety, the discriminator could easily distinguish its samples from the real ones, no?

P6:
Did you pass only the softmax values to the discriminator?

P7:
I like the score scheme introduced here. Do you see any connection to inception score?
So compared to normal GAN, does SIC-GAN have more parameters (due to the additional input)? If so, did you account for this in your experiments?

---

> ### Author Response · Authors · 2018-01-05
> **Thanks**
>
> We thank the reviewer for the positive comments. We have fixed the typos and grammar issues in the new version and cited more relevant work including the Bahdanau et al. 2016 “An actor-critic algorithm for sequence prediction” and the VAE RNN by Chung et al. 2015. Following is our reply to your specific comments:
>
> Q: Did you pass only the softmax values to the discriminator?
> A: No, we pass the mean and variance of each point generated by the Sketch RNN to the discriminator.
>
> Q: So compared to normal GAN, does SIC-GAN have more parameters (due to the additional input)?
> A: No, the parameter numbers of the unfolded Noisy GAN and SIG-GAN are roughly the same.
>
> Q: “Figure 2(d) displays human-drawn sketches of fire trucks which demonstrate that producing sequences–in this case sketches–with both quality and variety is definitely achievable in real-world applications”: I’m not sure I follow this argument. Because people can do it, ML should be able to?
> A: You are right. Here we just want to emphasize that the quality and variety is both achievable “by humans.” We have corrected the sentence in the paper.
>
> Q: Discriminator in MaliGAN/SeqGAN a simultaneous variety + quality reward signal?
> A: Yes it is. The MaliGAN/SeqGAN is proposed for the RNNs with the discrete output. While we use the continuous Sketch-RNN to demonstrate the Strong-Weak Collaborative GAN and SIC-GAN, the ideas could be readily applied to discrete cases. This is our future work.

---

### Decision · Program_Chairs · 2018-01-29
**ICLR 2018 Conference Acceptance Decision**

**Decision:**

Reject

**Comment:**

Pros and cons of the paper can be summarized as follows:

Pros:
* The underlying idea may be interesting
* Results are reasonably strong on the test set used

Cons:
* Testing on the single dataset indicates that the model may be of limited applicability
* As noted by reviewer 2, core parts of the paper are extremely difficult to understand, and the author response did little to assuage these concerns
* There is little mathematical notation, which compounds the problems of clarity

After reading the method section of the paper, I agree with reviewer 2: there are serious clarity issues here. As a result, I do cannot recommend that this paper be accepted to ICLR in its current form. I would suggest the authors define their method precisely in mathematical notation in a future submission.